# Food Security in a College Community: Assessing Availability, Access, and Consumption Patterns in a Mexican Context

**DOI:** 10.3390/ijerph22091314

**Published:** 2025-08-22

**Authors:** Wendy Jannette Ascencio-López, María Teresa Zayas-Pérez, Ricardo Munguía-Pérez, Guadalupe Virginia Nevárez-Moorillón, Manuel Huerta-Lara, María del Carmen Guadalupe Avelino-Flores, Teresa Soledad Cid-Pérez, Raúl Avila-Sosa

**Affiliations:** 1Posgrado en Ciencias Ambientales, Instituto de Ciencias, Benemérita Universidad Autónoma de Puebla, Cuidad Universitaria, Puebla 72570, Mexico; wendy.ascencio@alumno.buap.mx; 2Instituto de Ciencias, Benemérita Universidad Autónoma de Puebla, Cuidad Universitaria, Puebla 72570, Mexico; teresa.zayas@correo.buap.mx (M.T.Z.-P.); ricardo.munguia@correo.buap.mx (R.M.-P.); manuel.huerta@correo.buap.mx (M.H.-L.); 3Facultad de Ciencias Químicas, Universidad Autónoma de Chihuahua, Cuidad Universitaria, Chihuahua 31125, Mexico; vnevare@uach.mx; 4Facultad de Ingeniería Química, Benemérita Universidad Autónoma de Puebla, Cuidad Universitaria, Puebla 72570, Mexico; carmen.avelino@correo.buap.mx; 5Facultad de Ciencias Químicas, Benemérita Universidad Autónoma de Puebla, Cuidad Universitaria, Puebla 72570, Mexico; teresa.cid@correo.buap.mx

**Keywords:** food security, food access and availability, environmental health, higher education, dietary patterns, México

## Abstract

Food security among college students is an increasing concern, with potential implications for their health, academic performance, and future well-being. This study investigated food security within a college community in Mexico, focusing on food availability, access (both economic and physical), and consumption patterns. A mixed-methods approach was employed at Ciudad Universitaria, BUAP, Mexico, between 2023 and 2024. Stratified random sampling was used, resulting in a final sample of 606 students. Data were collected through structured questionnaires covering sociodemographic characteristics and eating habits, the ELCSA, structured cafeteria observations, semi-structured interviews with key informants, and three focus groups. Statistical analysis was performed using chi-square tests (*p* < 0.05). Post hoc analysis with Bonferroni adjustment confirmed that origin (*p* = 0.0017), mode of transportation (*p* = 2.31 × 10^−5^) and private vehicles (*p* = 1.77 × 10^−5^) were the key determinants. Although the environment offered a variety of options, processed and ultra-processed products dominated the food choices. A total of 95.9% of students purchased food on campus, yet only 21.8% reported engaging in healthy eating habits. Focus groups revealed that students’ food choices were influenced by availability, access, and perceptions of affordability and convenience. These findings highlight the urgent need for targeted interventions to improve food security and promote healthier dietary practices within the college setting.

## 1. Introduction

According to the Food and Agriculture Organization (FAO), food security is achieved when all people, at all times, have physical, social, and economic access to sufficient, safe, and nutritious food that meets their daily energy needs and dietary preferences for an active and healthy life [1]. Conversely, when individuals lack the capacity and mean to obtain such food for regular growth, development, and a healthy, active life, they experience food insecurity [2,3].

FAO analyzes food security through four dimensions: food availability; economic and physical access; consumption and biological utilization; and stability of supply [1,4]. While availability and access are necessary, they alone do not ensure adequate consumption. To promote food security, food must also be culturally acceptable, safe and socially appropriate [5,6]. Food acceptability—the cultural appropriateness of food—is recognized as a key element within the utilization dimension, as it facilitates both consumption and biological utilization [7].

Due to its multisectoral nature, food security manifests in various ways depending on the level of social organization being examined. However, given the inherent complexity of the phenomenon, along with methodological challenges, limitations in data availability, and the focus of some research, comprehensive assessment are not always possible. Consequently, many studies tend to adopt a unidimensional approach [8,9,10].

Unidimensional approaches, which analyze a single dimension of food security in isolation, are often employed to address specific research objectives, respond to public policy needs, or tackle urgent issues such as undernutrition or food access especially in contexts where these dimensions have an immediate impact on public health [9,11]. However, this approach has important limitations. By neglecting the interactions between the different dimensions, it provides a fragmented view and risks overlooking critical aspects such as the cultural context, local variations, and the synergy among the various factors influencing food security [12,13]. In contrast, multidimensional approaches, which integrate multiple dimensions simultaneously, enable a more holistic and comprehensive understanding of the determinants of food security [10,14]. This perspective supports more robust diagnoses and the development of more effective, relevant, and context-specific intervention strategies.

When applied to a specific context, analyzing college communities is particularly relevant, as the prevalence of food insecurity is higher among students than in the general population [15,16,17,18]. College students are a vulnerable group due to their adaptation to new environments, changes in schedules, academic workloads, and expenses [19]. Many students must relocate far from home, and some work part-time with limited or no income [3,20,21]. The lack of cooking facilities, culinary skills, and available time often prompts students to choose accessible, low-nutritional-value foods on campus—compromising their food security and fostering unhealthy practices. Even if food insecurity is temporary, this increases the risk of chronic diseases during their academic years [3,20,22].

Various studies have reported that the prevalence of food insecurity among college students. In Canada and South Africa, this phenomenon has been described as a “hidden threat” exposing students to greater risk than the general population [15]. In the United States ranges from 33% to 51% [16]; during the pandemic, Australia reported 32 to 48% [23]. In Europe, substantial rates are also documented: Germany (18.9–33%, many at moderate to severe levels) [3,24,25], Iceland (17%), Spain (19.6%), and Cyprus (14.9%) [26,27]. Lebanon shows the highest levels, 59–60.3%, underscoring the phenomenon’s geographical heterogeneity [3,25]. In Mexico, the incidence of food insecurity among college students is also high (14.6–59.5%), with prevalence rates of 16% for mild, 8% for moderate, and 5% for severe food insecurity, with sociodemographic factors such as low household educational level and recent work experience contributing to this issue [28].

Studies on food security highlight the college food environment as a critical factor influencing young people’s dietary habits. The food environment is defined as the physical, economic, political, and sociocultural context in which consumers interact with and experience the food system. Its components include factors such as availability, accessibility, and convenience, among others [29,30].

Availability is a key dimension of food security in colleges [29]. It involves ensuring sufficient, diverse, and high-quality food [2], primarily depending on the location of food outlets and market supply. While location and availability alone do not determine access, their scarcity—and the increase availability of accessible, unhealthy fast food—can contribute to food insecurity and poor dietary habits [31,32].

Access refers to how individuals acquire the food they consume [33], and can be analyzed from physical, economic, and social perspectives. Economic access depends on individuals’ ability and capacity to obtain food, which is influenced by factors such as purchasing power and available social support [2,33]. Purchasing power is determined by the relationship between per capita income, household income, and the cost of food. Physical access relates to the infrastructure that enables individuals to reach and obtain food [33].

Food consumption is a fundamental component of food security, influenced by availability, access, and acceptability at the individual, family, and community levels [34,35,36]. Acceptability pertains to both consumption and biological utilization, referring to the population’s ability to make appropriate decisions about selecting, storing, preparing, distributing, and consuming food. These decisions are shaped by environment factors such as customs, dietary practices, educational level, media, and food technologies [2,37].

Despite the increasing interest in food insecurity among university students, the literature reveals significant theoretical and methodological gaps. Most studies focus exclusively on economic access to food, neglecting the other dimensions outlined by the FAO. Only a limited number of studies address these dimensions simultaneously [6,9]. Additionally, there is still little exploration of how food insecurity impacts health, as well as limited evaluation of effective institutional strategies within the college setting [38]. In contexts such as Mexico, where structural factors like poverty, marginalization and environmental vulnerability directly affect food access and consumption, these gaps are particularly salient [39].

In this regard, an integrated perspective entails simultaneous and coherently considering all four components—availability, access, acceptability, and consumption—aiming to understand how they interact to influence food security. Although some studies have addressed this multidimensional problem in other populations [40,41,42,43], there remains limited evidence that comprehensively integrates these dimensions within the college setting. This underscores the importance contribution of this work. This study aims to provide a comprehensively evaluation of food availability, access, acceptability, and consumption as key factors influencing the current prevalence of food insecurity among college students.

## 2. Materials and Methods

This descriptive–analytical cross-sectional study was conducted at the Ciudad Universitaria campus of the Benemérita Universidad Autónoma de Puebla (BUAP), Mexico. Data were collected between 8 March 2023, and 30 April 2024, employing a mixed methods design, integrating both quantitative and qualitative approaches.

### 2.1. Population and Sample

The study population comprised students enrolled in undergraduate and graduate programs at Ciudad Universitaria, BUAP, as well as managers of food outlets located within the campus.

For the quantitative component, a stratified random sampling method proportional to size of each stratum was applied based on the 2023 Yearbook, categorized by faculty and educational program. The initial sample size, calculated for a finite population, with 10% margin of error and 95% confidence level, was 381 students. To improve representativeness, the sample was increased to 606 students.

Inclusion criteria required enrollment, age ≥ 18, and informed consent with guaranteed confidentiality. Participants submitting incomplete or inconsistent questionnaires or withdrawing consent were excluded.

For the qualitative component, three focus groups, each comprising six students (total *n* = 18), were purposively sampled [44]. Participants were selected based on academic affiliation, gender balance and willingness to participate, to ensure diverse perspectives on the university’s food environment. Students from faculties such as Chemical Sciences, Engineering, Biological Sciences, Physical Culture, Law, Administration, and Architecture participated. Additionally, managers of food establishments within the campus were included through direct contact at their respective venues.

### 2.2. Quantitative Variables, Instruments, and Data Collection

#### 2.2.1. Food Security Status

This variable was categorized into four levels: food security, mild food insecurity, moderate food insecurity, and severe food insecurity. Instrument: Latin American and Caribbean Food Security Scale (ELCSA), a validated tool for assessing dimensions of food access through multiple indicators [45]. Instruments were selected based on previous studies in college populations and reviewed by public health experts to ensure content validity.

#### 2.2.2. Sociodemographic and Economic Variables

These included age, sex, origin (local/non-local), faculty, academic level, employment status (employed/not employed), housing type (owned, rented, shared, university residence, or other), means of transportation used (public transportation, personal vehicle, bicycle, walking or other) and type of economic dependence (dependent or independent). Parental education levels, number of household income earners, access to food grants (yes/no), and weekly food expenditure were also recorded. Instrument: Structured Questionnaire—Sociodemographic and Economic Information.

#### 2.2.3. Acceptability, Eating Habits, and Behaviors

Variables included satisfaction with campus food supply (Likert scale: very dissatisfied–very satisfied), consumption preferences (on-campus consumption, off-campus, bringing food from home) and frequency of cafeteria use (always, almost always, sometimes, almost never, never). Eating habits and behavior s were measured via ordinal Likert scales, assessing label reading, expiration date checks, weekly frequency of consuming fast foods, sweets, and snacks; number of daily meals; weekly consumption of legumes, vegetables, fruits, fish, and cereals; daily hours spent sedentarily; food consumption between meals and frequency of eating without hunger; eating while engaged in other activities; and consumption of ultra-processed foods. Additionally, weekly intake of pork, dairy products, bread and pasta was recorded. Based on composite scores, habits were classified as deficient (<2.74, below the 25th percentile), sufficient (2.75–3.10, between the 25th and 75th percentiles), or healthy (>3.22, above the 75th percentile).

Instrument: Structured Questionnaire—Consumption Preferences and Satisfaction Levels and Eating Behaviors and Habits, previously validated through factor analysis and Cronbach’s alpha by Díaz et al. [46] and adapted by public health experts for this study. Included eight sections with a total of 19 Likert scale questions covering themes such as Food Concern, Fast Food Consumption, Dietary Guidelines, Sedentary Eating Behavior, Eating Outside Regular Hours, Consumption of Unnecessary Foods, High-Consumption Foods, and Satiety with Energy-Dense Foods. Habit classification was based on the average score for each student, compared to the maximum possible score.

### 2.3. Qualitative Variables, Instruments, and Data Collection

#### 2.3.1. Availability and Physical Access to Food

Variables encompassed the availability and variety of foods sold in campus cafeterias (supply and demand), classification according to the NOVA system into unprocessed/minimally processed, processed, ultra-processed, and prepared foods (the latter defined as combinations of minimally processed foods with culinary ingredients or processed products based on NOVA criteria), and physical access to food based on available infrastructure such as the presence of pubs, consumption areas, and points of sale.

Instruments: Structured Observation of the Food Environment (37 cafeterias) and Semi-structured Interviews with Cafeteria Managers (*n* = 37). The interview guide was developed based on literature concerning food environments and food supply criteria in college and school settings, following methodological recommendations for qualitative semi-structured interviews [47,48,49,50]. A pilot test was conducted with managers excluded from the final sample. Interviews were held at the establishments themselves, each lasting approximately 25 min, with informed consent obtained prior to participation.

#### 2.3.2. Focus Group Analysis of Perceptions, Habits, and Experiences

Variables emerged from thematic analysis of focus groups and interviews, including perceptions of the campus food supply, factors influencing consumption decisions (price, taste, satiety, custom, practicality), barriers to accessing healthy options (economic, logistical, cultural), satisfaction and suggestions for improvement, as well as the influence of social norms, advertising and family habits.

Instrument: Focus Group Guide (semi-structured discussion protocol). Sessions were conducted face-to-face between March and April 2024, in college spaces, lasting approximately 90 min. All sessions were transcribed with prior authorization, respecting ethical principles. To ensure the credibility of qualitative analysis, techniques such as triangulation, cross-coding by researchers, and selection of representative fragments were employed.

### 2.4. Data Analysis

Quantitative data were analyzed using Excel (version 2309) and the R-Studio. Descriptive statistics were calculated, and normality was assessed with the Shapiro–Wilk test, which resulted in a *p*-value of 0.048. The mean score was 2.91 (SD = 0.29), indicating low variability within the data. To examine the association between food insecurity and a range of socioeconomic and demographic variables, Pearson’s Chi-square test was employed, suitable for the categorical nature of these variables. The post hoc analysis was carried out using standardized residuals adjusted with Bonferroni correction.

For qualitative analysis, an inductive thematic approach was used to process the interviews and focus groups transcripts. Two researchers independently coded the transcripts, with cross-checking and consensus-building to ensure reliability. The units of meaning were organized into emergent categories and subcategories, which were then integrated into an interpretive framework. A methodological triangulation strategy was applied to combine quantitative and qualitative findings, contrasting statistical patterns with participants’ experiences and perceptions to facilitate a more contextualized understanding of the studied phenomenon.

### 2.5. Ethical Aspects

The study was conducted in accordance with the principles of the Declaration of Helsinki and in line with current national regulations. All participants were informed about the study objectives, voluntary participation, and confidentiality. Written or verbal informed consent was obtained prior to data collection. Interviews and focus groups were recorded only after obtaining explicit consent from participants.

## 3. Results

### 3.1. Food Security Status

The results from the application of the Latin American and Caribbean Food Security Scale (ELCSA) indicate that 45.2% of the community is food secure, while 54.8% experience food insecurity. Among those facing insecurity, 28.4% were classified as having mild insecurity, 19.0% as moderate, and 7.4% as severe. Limited access to food, which is a key indicator of food security according to Mexican Consejo Nacional de Evaluación de la Política de Desarrollo Social (CONEVAL), encompasses cases of moderate and severe food insecurity. In this study, such cases were observed in 26.4% of the evaluated population, highlighting a significant proportion in a state of nutritional vulnerability.

### 3.2. Sociodemographic Characteristics and Economic Access

The average age of participants was 20.1 years (SD = 1.9), with 52.6% identifying as female. Sociodemographic characteristics by household food insecurity levels are presented in Table 1.

The sample was diverse in terms of origin, with 41.4% being non-local students and 58.6% local students. Regarding employment, 34.8% of the students reported working, and 80.4% primarily use public transportation for commuting. Concern access to institutional support, 97.5% reported not receiving a food scholarship. Additionally, 92.6% of the students are economically dependent on third parties, with 72.6% living with their parents. Most households had two working members (41.1%). In terms of weekly food expenditure, 64.9% reported a budget between $5.00 and $30.00 USD, while 2.0% indicated no expenses in this category, as they preferred to bring their own food from home.

The educational levels of the father and mother were predominantly college level, with 38.8% and 33.7%, respectively. Regarding their occupations, it was observed that 86.8% of fathers and 59.6% of mothers were employed; however, many fathers were retired (6.6%), absent due to not living at home or are deceased (4.8%). Among mothers, 36.5% dedicated themselves to household chores, while 4.0% are retired, pensioned, or deceased.

Chi-square tests revealed significant associations between food security status and origin, employment status, transportation mode, economic dependence, household composition, and father’s education level. Local students exhibited higher proportions of food security compared to non-local students; however, paradoxically, they also exhibited higher rates of severe food insecurity (*p* = 0.0001). Students who worked had a higher prevalence of severe food insecurity (11.4%) than those who did not work (5.3%, *p* = 0.003). Transportation mode was strongly linked to food security (*p* < 0.0001), students with their own vehicle reported greater food security (73.3%).

Regarding household composition, students living with their parents experienced greater food security (50.0%), while those living alone, with relatives other than parents, or with partners showed higher levels of food insecurity (*p* = 0.001). Additionally, students whose parents completed university studies (50.2%) or postgraduate education (52.0%) experienced greater food security, whereas moderate (33.3%) and severe (26.7%) food insecurity were associated with lack of formal education (*p* = 0.022). Although the mother’s educational level was not statistically significant (*p* = 0.628), similar patterns emerged: higher food security among those with mothers having university (49.5%) or graduate studies (51.4%).

Post hoc analysis with Bonferroni-adjusted residuals confirmed that origin (adjusted α = 0.0063, *p* = 0.0017) and transportation mode (adjusted *p* < 0.0042) were the only category-level associations remaining statistically significant. Specifically, local students and those using private vehicles demonstrated higher food security, while non-locals and students relying on public transport, walking, or cycling faced higher insecurity. No significant association were found within employment status, economic dependency, household composition, or father’s education level after adjustment, indicating that these relationships reflect overall trends rather than effects driven by specific subgroups.

Economic accessibility of food for students relates to the balance between disposable income and food costs. In this study, most students reported that the food available in campus cafeterias was generally affordable, considering their limited budgets. Additionally, some cafeterias provided options to heat food brought from home for a minimal fee.

### 3.3. Availability and Access to Food

The food environment at Ciudad Universitaria BUAP includes various points of sale, such as vending machines, the University Pavilion, and 37 food cafeterias distributed across the 14 faculties. This proximity to food sources facilitates physical access to a wide range of food options during students’ academic day. However, some faculties were identified to have insufficient infrastructure for food consumption, such as lack of tables and shaded areas like palapas. The analysis of the food supply revealed a broad diversity of available products, including breakfast options, full meals, à la carte dishes, as well as snacks and appetizers intended for consumption between meals. Interviews conducted with the managers of these food outlets provided insights into the factors influencing the food supply, which are illustrated in Figure 1.

The primary determinant influencing food supply was student demand, as consumer preferences directly affect the availability of food and beverages. Similarly, economic access—both for the managers, who consider sourcing costs, and for the students, based on product prices—was a key factor in decision-making. Another relevant factor was the diversity of offerings; a broader variety of products was viewed as increasing sales opportunities. Interviewees also expressed interest in offering healthy food options, although this was not always prioritized in product selection. Less frequently, managers highlighted practicality as a criterion, favoring ingredients that can be used in multiple preparations and enable quick preparation, including both ready-to-eat and ready-to-sell products. Additional considerations include taste, satiety, perceived quality, product innovation, and the availability of basic and traditional options. Other criteria involved preferences for foods and beverages that do not require heating, the ratio of quantity to price, and the inclusion of tortilla-based products or items easily prepared on-site. Notably, only one establishment offered specific products for vegetarian or vegan consumers.

An analysis of the classification of available foods revealed a wide diversity that reflects the heterogeneity of student preferences. Based on direct observation and interview data, it was observed that the most commonly offered and consumed products were prepared foods—such as chilaquiles, tortas, enchiladas, baguettes, tacos, sandwiches, tostadas and desserts such as pie, flan and rice pudding. These were followed by ultra-processed foods, including pizzas, hamburgers, hot dogs, French fries, chocolates, energy bars, cookies, fried snacks, candy, soft drinks and flavored beverages. In contrast, fresh or minimally processed foods—such as salads, fresh fruit and natural juices—had lower availability and demand. Plain water was identified as the beverage with the highest consumption.

### 3.4. Acceptability and Food Consumption

Although 73.6% prefer to consume food at home, 95.9% purchase food from campus cafeterias. Only 22.5% expressed consistent satisfaction with the variety of available food, while 37.2% were dissatisfied and 40.3% held a neutral opinion. Perceptions of satisfaction or dissatisfaction were influenced by factors such as product variety, price, availability, and alignment with personal preferences. Additionally, some students expressed negative perceptions of the nutritional value of campus food and preferred to bring their own food from home.

The overall assessment of dietary behaviors and habits indicated that 48.0% of respondents practiced sufficient eating habits, 30.2% exhibited deficient habits (Table 2), and only 21.8% demonstrated healthy eating practices.

Moreover, 32.7% of students almost never check food labels. Only 42.1% consume at least one portion of fruit daily, and 33.7% one portion of vegetables. Additionally, 52.6% do not consume fish, 34.0% eat meat three times a week, and 44.8% do not consume cereals. Sedentary behavior is high: 55.1% spend between 4 and 6 h sitting daily, while 35.8% spend between 7 and 9 h. Furthermore, 63.6% eat while watching TV, using their phones, or reading. Although 48.8% have three meals a day, 32.5% only have two.

### 3.5. Focus Groups

The results revealed key factors influencing food decisions, including availability, physical and economic access, and nutritional knowledge (Table 3). Social factors—such as beliefs and myths influenced by advertising, food trends, and social norms—also play an important role.

During the discussions, participants highlighted that the low consumption of healthy options was due to their limited availability, high costs, and a lack of awareness of their benefits. The absence of promotional strategies and informational campaigns further exacerbates the situation. Participants proposed incorporating functional foods and encouraging student involvement in projects related to healthy eating and food hygiene as strategies to prevent diseases.

## 4. Discussion

This study conducts an analysis comparing key dimensions of the food environment in colleges. The findings indicated that the prevalence of food insecurity (54.8%) exceeds the values reported in similar contexts. In the United States, the average prevalence ranges between 33% and 51% [16,51,52], with figures comparable to those reported in Australia (32% and 48%) [23]. A systematic review of studies across the USA, Australia, Canada, South Africa, and Malaysia estimated an average prevalence of 42%, with a range of 12.5% to 84% [53]. Conversely, lower rates have been documented in European countries such as Germany (18.9–33%) [3,24,25], Iceland (17%), Spain (19.6%) and Cyprus (14.9%) [26,27]. According to Manikas et al. [6], students of Hispanic origin are more susceptible to food insecurity, which may partly explain regional differences. Countries such as Lebanon report notably higher rates, ranging from 59% to 60.3% [3,25].

In Mexico, 30.8% of university students experienced food insecurity in 2018, with a value of 31.4% reported for Puebla [28], a figure substantially lower than that observed in our study, potentially due to regional, methodological, or temporal factors. Internationally, evidence suggests that university students often exhibit higher levels of food insecurity compared to the general population [52]. However, the prevalence in our study is slightly below the 55.5% recorded in the Mexican general population in 2018–2019, prior to the COVID-19 pandemic [54]. By 2021, this figure rose to 60.8%, attributed to the socioeconomic impacts of the pandemic [55]. These findings indicate that, although university students are a vulnerable group, their food security status may be influenced by specific academic or regional factors that merit further investigation.

Regarding sociodemographic factors, post hoc analysis with Bonferroni correction identified origin and transportation mode as key determinants of food security. Consistent with prior studies linking food security to place of residence [56], and reliance on public transport [57,58], local students demonstrated higher overall food security but paradoxically showed greater rates of severe insecurity, indicating vulnerable subgroups within the local population that merit targeted interventions. The strong association between transportation mode and food security emphasizes logistical barriers: students relying on public transportation face substantial challenges in accessing healthy food, which exacerbates their vulnerability. While employment status, economic dependence, household composition, and father’s educational level showed overall associations with food insecurity, these were not statistically significant after Post Hoc category-specific analysis. Nonetheless, previous research indicates that financial dependence influences food security, highlighting the importance of family networks as buffers against food insecurity [57,58]. Evidence also suggests that employed students are at greater risk of food insecurity, whereas being unemployed is negatively associated with it [28]. The link between food insecurity and low paternal educational attainment aligns with existing literature that identifies parental education as a structural determinant that perpetuates inequities in access to adequate food [59,60]. Additionally, postgraduate students have been shown to exhibit higher food security compared to undergraduate students [61]. It is noteworthy that only 2.5% of the students receive a food scholarship, which likely limits its impact at the population level and explains the lack of significant association. Therefore, further research is needed to evaluate the program’s efficacy and the criteria used for its allocation

The diversity and location of food outlets enhance their proximity, thereby facilitating availability and physical access to various food options. However, in some faculty’s insufficient infrastructure for food consumption—such as lack of tables and suitable eating spaces—was identified. Additionally, there is a predominance of ultra-processed products and meals prepared from these products. This pattern aligns with findings from numerous countries, which have documented a widespread supply of foods and beverages high in calories and low in nutritional value [62,63]. This trend is consistent with the increase in processed and ultra-processed foods in Latin America documented by the Pan American Health Organization [64], and poses a significant challenge for food security. Although physical access and availability are generally adequate, these conditions do not guarantee healthy eating, nor do they directly contribute to improving food security. As previous research indicates [65,66], unhealthy food environments tend to promote poor dietary behaviors and increase metabolic risk. Conversely, the availability and accessibility of healthy options support healthier food choices [66,67]. This highlights the need to strengthen certification criteria for healthy colleges.

Regarding food acceptability and consumption, it was observed that food decisions are influenced by multiple interrelated factors, consistent with the understanding that environmental, social, and individual determinants interact complexly [21,66]. The food environment itself is favored by availability—a key element according to Li et al. [29]—as well as the variety of products offered and the physical and economic access to these foods. Sociocultural factors, such as beliefs, myths, social norms, trends, and advertising, also significantly influence eating habits. They emphasize that parents, friends, and media play vital roles in shaping consumption patterns. Finally, among individual determinants, the alignment of food choices with personal preferences—particularly perceptions of satisfaction—was crucial. Numerous studies have identified taste as the primary food motivation for food selection, followed by cost and convenience [66,67,68].

Although sufficient eating behaviors were observed quantitatively, several inadequate practices were also identified in line with the Dietary Guidelines for the Mexican Population [69], such as low consumption of fruits, vegetables, and fish, skipping main meals, limited interest in nutritional labels, and a sedentary lifestyle. These findings are consistent with studies documenting unhealthy eating patterns among university students, characterized by high intake of ultra-processed foods and fast food [70], limited nutritional knowledge, infrequent label reading, and low participation in educational programs [15,20,71]. These factors contribute to energy imbalances and increase the risk of chronic diseases.

This evidence enriches to the scientific debate by moving beyond reductionist approaches to food security, providing a solid foundation for interventions that acknowledge the sociocultural complexity of food security within the college population. From the neoclassical consumer choice perspective, students’ food decision-making is explained by budget constraints and utility maximization, where cost, limited income, and preferences influence the selection of foods, including ultra-processed options [72,73]. Simultaneously, the study aligns with Douglas and Wildavsky’s cultural theory of food and risk, which posits that food practices are shaped by symbolic rules, social norms, and the construction of collective identities [74,75].

Although this study was conducted at a single public institution, its application to a university with high enrollment and regional relevance offers valuable insights into a population that is often underrepresented in scientific literature. A key strength lies in its comprehensive methodological approach, combining analysis of the food environment with sociodemographic variables and individual perceptions of consumption. This allows for a broader, more contextualized understanding of the determinants of food security in the university setting, addressing the gap of tools that currently consider these dimensions in student communities. The use of validated scales, such as the ELCSA, alongside the examination of less-explored factors like type of coexistence, transportation means and parental education level, further enhances the relevance and applicability of the findings.

Among the limitations, it is acknowledged that aspects such as the availability, variety and predominance of products were assessed though observational methods, which limited detailed quantitative measurements of price, nutritional quality, and supply stability. While this approach offers a valuable overview, it restricts in-depth analysis of the economic and nutritional barriers faced by students. Additionally, the cross-sectional design limits the ability to establish causal relationships among the variables studied. Future research should incorporate multivariate statistical analyses and longitudinal designs to develop more robust explanatory models suitable for public policy development. Overall, this study provides relevant evidence to inform institutional decision-makers, policymakers and program designers aiming to improve food conditions within the university environment.

## 5. Conclusions

This study reveals a high prevalence of food insecurity among college students, initially associated with origin, employment status, mode of transportation, economic dependence, household composition, and the father’s educational level. However, after Bonferroni-adjusted post hoc analysis, only origin and mode of transportation remained significantly associated, highlighting their crucial role as key structural determinants within this population. Although there is generally sufficient availability and physical access to meet the average energy requirements, the limited supply of healthy foods may alter dietary patterns, impact well-being, and influence the dietary choices of students. These findings provide policymakers with valuable insights to assess the severity of the issue and develop targeted strategies to reduce its prevalence and promote better health outcomes among students.

## Figures and Tables

**Figure 1 ijerph-22-01314-f001:**
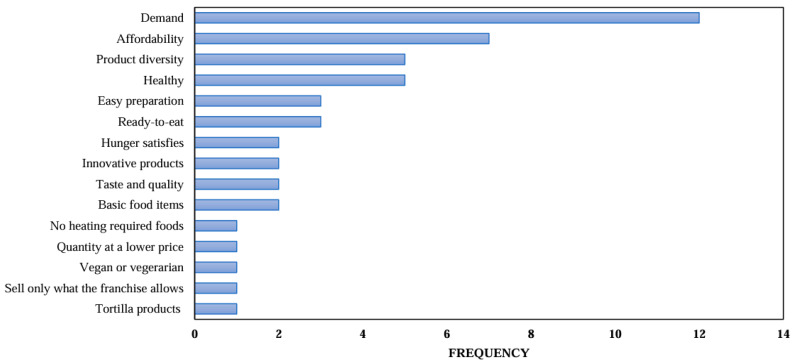
Determinants of food supply in Ciudad Universitaria BUAP food outlets, México, n = 37, 2023.

**Table 1 ijerph-22-01314-t001:** Sociodemographic characteristics in college students at Ciudad Universitaria BUAP, México (*n* = 606), 2023.

College Student’s Sociodemographic Characteristics	Total *n* (%)	Food Security *n* = 274 (45.2%)	Mild FI *n* = 172 (28.4)	Moderate FI *n* = 115 (19.0%)	Severe FI *n* = 45 (7.4%)	*p*-Value ^2^
Origin						0.0001
Non-local	251 (41.4)	90 (35.9) ^a^	85 (33.9)	61 (24.3)	15 (5.9)	
Local	355 (58.6)	184 (51.8) ^a^	87 (24.5)	54 (15.2)	30 (8.5)	
Employment						0.003
No	395 (65.2)	194 (49.1)	114 (28.9)	66 (16.7)	21 (5.3)	
Yes	211 (34.8)	80 (37.9)	58 (27.5)	49 (23.2)	24 (11.4)	
Transportation						<0.0001
Public	487 (80.4)	193 (39.7) ^b^	156 (32.0)	96 (19.7)	42 (8.6)	
Private vehicle	86 (14.2)	63 (73.3) ^c^	11 (12.8)	10 (11.6)	2 (2.3)	
Bicycle/Walking	33 (5.4)	18 (54.5)	5 (15.2)	9 (27.3)	1 (3.0)	
Food Scholarship						0.156
No	591 (97.5)	269 (45.5)	168 (28.4)	109 (18.5)	45 (7.6)	
Yes	15 (2.5)	5 (33.3)	4 (26.7)	6 (40.0)	0 (0.0)	
Financial Dependence						0.009
Yes	561 (92.6)	263 (46.9)	158 (28.2)	102 (18.1)	38 (6.8)	
No	45 (7.4)	11 (24.4)	14 (31.1)	13 (28.9)	7 (15.6)	
Living Arrangement						0.001
Parents	440 (72.6)	220 (50.0)	124 (28.2)	72 (16.4)	24 (5.4)	
Relatives	79 (13.0)	27 (34.2)	19 (24.0)	22 (27.9)	11 (13.9)	
Roommates	26 (4.3)	6 (23.1)	9 (34.6)	9 (34.6)	2 (7.7)	
Alone	61 (10.1)	21 (34.4)	20 (32.8)	12 (19.7)	8 (13.1)	
Father’s Education						0.022
Graduate degree	25 (4.1)	13 (52.0)	6 (24.0)	5 (20.0)	1 (4.0)	
Undergraduate degree	235 (38.8)	118 (50.2)	62 (26.4)	41 (17.5)	14 (5.9)	
High school	174 (28.7)	84 (48.3)	46 (26.4)	30 (17.2)	14 (8.1)	
Junior high school	108 (17.8)	40 (37.0)	41 (38.0)	22 (20.4)	5 (4.6)	
Elementary school	49 (8.1)	16 (32.6)	14 (28.6)	12 (24.5)	7 (14.3)	
No education	15 (2.5)	3 (20.0)	3 (20.0)	5 (33.3)	4 (26.7)	
Mother’s Education						0.628
Undergraduate degree	35 (5.8)	18 (51.4)	11 (31.4)	2 (5.7)	4 (11.5)	
High school	204 (33.7)	101 (49.5)	56 (27.4)	33 (16.2)	14 (6.9)	
Junior high school	170 (28.1)	77 (45.3)	49 (28.8)	30 (17.7)	14 (8.2)	
Elementary school	141 (23.3)	59 (41.8)	40 (28.4)	33 (23.4)	9 (6.4)	
Undergraduate degree	41 (6.8)	18 (43.9)	13 (31.7)	7 (17.1)	3 (7.3)	
No education	15 (2.5)	3 (20.0)	6 (40.0)	4 (26.7)	2 (13.3)	
Father’s Occupation						0.248
Employed	526 (86.8)	243 (46.2)	153 (29.1)	94 (17.9)	36 (6.8)	
Unemployed	49 (8.1)	21 (42.9)	9 (18.4)	13 (26.5)	6 (12.2)	
Other	31 (5.1)	10 (32.3)	10 (32.3)	8 (25.8)	3 (9.6)	
Mother’s Occupation						0.243
Employed	361 (59.6)	168 (46.6)	90 (24.9)	74 (20.5)	29 (8.0)	
Homemaker	221 (36.5)	95 (43.0)	74 (33.5)	39 (17.6)	13 (5.9)	
Other	24 (4.0)	11 (45.9)	8 (33.3)	2 (8.3)	3 (12.5)	

*p*-value < 0.05 was considered statistically significant. ^2^ Data are presented as frequency (percentage). ^a^ Significant association (*p =* 0.0017), after Bonferroni correction (adjusted α = 0.0063). ^b^ Significant association (*p =* 2.31 × 10^−5^) after Bonferroni correction (adjusted α = 0.0042). ^c^ Significant association (*p =* 1.77 × 10^−5^) after Bonferroni correction (adjusted α = 0.0042).

**Table 2 ijerph-22-01314-t002:** Eating behaviors and habits categories in college students at Ciudad Universitaria BUAP, México (*n* = 606), 2023.

Categories	Freq.	Percent
Healthy	132	21.78
Sufficient	291	48.02
Deficient	183	30.20

Results based on the overall average of responses obtained from students. Score >3.11 Healthy, 2.75–3.10 Sufficient, <2.74 Deficient.

**Table 3 ijerph-22-01314-t003:** Themes and subthemes associated with the campus food environment, consumption options, and food insecurity in college students at Ciudad Universitaria BUAP, México (n = 18), 2023.

Theme	Subtheme	Illustrative Quotes
Access and Availability	Main barriers to purchasing	“I think food prices can be high.” “I only eat at cafeterias near me because I have little time to eat.” “I prefer set meals because they are more complete, but there are hardly any cafeterias that offer them.” “I frequently use the water fountain so I don’t spend money on buying water.” “My class schedule doesn’t allow me to eat properly.”
Food Acceptability	Variety of consumption	“I prefer eating at a variety of cafeterias to explore other options.”
Healthy Food Options	Types of foods Advertising Food safety	“They could offer functional foods and take advantage of student projects so we can participate in the proposals.” “The cafeterias don’t have advertising that promotes healthy eating.” “They should have campaigns on healthy eating through classroom screens or other methods.” “Yes, I’ve felt sick to my stomach after eating at some cafeterias.”

## Data Availability

All the data generated or analyzed during this study are included in this published article.

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
