# Peer review of "Food Security in a College Community: Assessing Availability, Access, and Consumption Patterns in a Mexican Context"

_ijerph, 2025, doi:10.3390/ijerph22091314_

Round 1
Reviewer 1 Report
Comments and Suggestions for Authors
The present manuscript proffers a comprehensive mixed-methods study investigating food security within a Mexican university setting. The study focuses on food availability, access, consumption patterns, and sociodemographic associations. The study is timely, methodologically robust, and contributes significantly to the understanding of food insecurity among university students in Latin America. However, some areas require further elucidation and refinement in order to enhance the manuscript's clarity, rigor, and overall impact. The following is a detailed outline of the aforementioned points.
- The abstract section is quite insufficient and does not fully express the manuscript. The findings of the study should also be briefly included in this section. The abstract lists "Food security; food access and availability; environmental health; higher education; dietary patterns." Consider adding "Mexico" or "Mexican context" to the keywords to enhance searchability and specificity.
- While the introduction delineates the dimensions of food security (availability, access, consumption, utilization and stability), the study's stated objective (Line 96) encompasses "acceptability." The ensuing discourse in the methods and results sections primarily engages with the concept of "acceptability" within the context of consumption preferences and satisfaction. It would be beneficial to explicitly define "acceptability" as a distinct dimension or to clarify its relationship to "consumption and biological utilization" as per the Food and Agriculture Organization's (FAO) framework outlined in the introduction, and to consistently apply this definition throughout the text.
- It is notable that the p-values in Table 1 are reported as "<0.001" for a number of variables. While this indicates a high level of significance, the provision of the exact p-value (e.g. p=0.000) or a more precise range (e.g. p<0.0001) would be more informative for readers, especially given the use of R-Studio, which typically provides exact values.
- The study observes that merely 2.5% of students are beneficiaries of the food scholarship (Lines 238-240). While the conclusion mentions "limited program coverage and the need to evaluate its reach and allocation criteria," the discussion (lines 313-317) states that "the food scholarship... did not show significant associations, suggesting that while these factors have been considered relevant socioeconomic determinants in other contexts, their influence may be mediated by contextual elements." This appears to be a contradictory statement. Should the benefit be shown to be only 2.5%, the impact on the overall student body's food security status would naturally be minimal. This is different from the result not showing a "significant association" in a statistical sense. It is imperative that this issue is clarified during the course of the discussion. In order to achieve this, emphasis should be placed on the fact that the limited reach of the scholarship in question inherently limits its observable statistical association with food security across the entire sample.
- The authors state that "there are currently no studies analyzing this issue from an integrated perspective" (Lines 97-99). Whilst the employment of a mixed-methods approach is to be commended, the discussion makes extensive comparisons between the findings and those of other studies that also examine multiple dimensions (e.g. environmental, social, individual determinants). Furthermore, the literature contains many valuable studies of similar types, both in Mexico and elsewhere. It is a shortcoming that the manuscript does not cite many of these studies, some of which are presented below.
- Appendini, K., & Quijada, M. G. (2016). Consumption strategies in Mexican rural households: pursuing food security with quality. Agriculture and Human Values, 33, 439-454.
- Calloway, E. E., Carpenter, L. R., Gargano, T., Sharp, J. L., & Yaroch, A. L. (2023). New measures to assess the “Other” three pillars of food security–availability, utilization, and stability. International Journal of Behavioral Nutrition and Physical Activity, 20(1), 51.
- Cruz-Sánchez, Y., Aguilar-Estrada, A., Baca-del Moral, J., & Monterroso-Rivas, A. I. (2024). The availability of food in Mexico: an approach to measuring food security. Agriculture & Food Security, 13(1), 35.
- Martínez-Martínez, O. A., Gil-Vasquez, K., & Romero-González, M. B. (2023). Food insecurity and levels of marginalization: food accessibility, consumption and concern in Mexico. International Journal for Equity in Health, 22(1), 178.
- Line 132: Please update the reference "[NO_PRINTED_FORM]" to the correct citation for the questionnaire validation.
- Line 298: "On the other hand, although physical access and availability are adequate, they do not guarantee healthy eating or better food security status..." This sentence is a bit long. Consider breaking it into two or rephrasing for better flow.
The quality of the English language throughout the text is generally good, but could be improved with a few corrections, which I have included in the comments.
Author Response
REVIEWER 1
The present manuscript proffers a comprehensive mixed-methods study investigating food security within a Mexican university setting. The study focuses on food availability, access, consumption patterns, and sociodemographic associations. The study is timely, methodologically robust, and contributes significantly to the understanding of food insecurity among university students in Latin America. However, some areas require further elucidation and refinement in order to enhance the manuscript's clarity, rigor, and overall impact. The following is a detailed outline of the aforementioned points.
- The abstract section is quite insufficient and does not fully express the manuscript. The findings of the study should also be briefly included in this section. The abstract lists "Food security; food access and availability; environmental health; higher education; dietary patterns." Consider adding "Mexico" or "Mexican context" to the keywords to enhance searchability and specificity.
RESPONSE: We thank you for your observation. The visible changes have been added in lines: 23-34, 40.
- While the introduction delineates the dimensions of food security (availability, access, consumption, utilization and stability), the study's stated objective (Line 96) encompasses "acceptability." The ensuing discourse in the methods and results sections primarily engages with the concept of "acceptability" within the context of consumption preferences and satisfaction. It would be beneficial to explicitly define "acceptability" as a distinct dimension or to clarify its relationship to "consumption and biological utilization" as per the Food and Agriculture Organization's (FAO) framework outlined in the introduction, and to consistently apply this definition throughout the text.
RESPONSE:The visible changes have been added in lines: 53-57.
- It is notable that the p-values in Table 1 are reported as "<0.001" for a number of variables. While this indicates a high level of significance, the provision of the exact p-value (e.g. p=0.000) or a more precise range (e.g. p<0.0001) would be more informative for readers, especially given the use of R-Studio, which typically provides exact values.
RESPONSE:We appreciate your comment. In the revised version of the manuscript, they are presented more accurately:
- For origin, p = 0.0001.
- For means of transport, p < 0.0001.
These changes were already reflected in the new version of Table 1.
- The study observes that merely 2.5% of students are beneficiaries of the food scholarship (Lines 238-240). While the conclusion mentions "limited program coverage and the need to evaluate its reach and allocation criteria," the discussion (lines 313-317) states that "the food scholarship... did not show significant associations, suggesting that while these factors have been considered relevant socioeconomic determinants in other contexts, their influence may be mediated by contextual elements." This appears to be a contradictory statement. Should the benefit be shown to be only 2.5%, the impact on the overall student body's food security status would naturally be minimal. This is different from the result not showing a "significant association" in a statistical sense. It is imperative that this issue is clarified during the course of the discussion. In order to achieve this, emphasis should be placed on the fact that the limited reach of the scholarship in question inherently limits its observable statistical association with food security across the entire sample.
RESPONSE:The visible changes have been added in lines: 451-455
- The authors state that "there are currently no studies analyzing this issue from an integrated perspective" (Lines 97-99). Whilst the employment of a mixed-methods approach is to be commended, the discussion makes extensive comparisons between the findings and those of other studies that also examine multiple dimensions (e.g. environmental, social, individual determinants). Furthermore, the literature contains many valuable studies of similar types, both in Mexico and elsewhere. It is a shortcoming that the manuscript does not cite many of these studies, some of which are presented below.
- Appendini, K., & Quijada, M. G. (2016). Consumption strategies in Mexican rural households: pursuing food security with quality. Agriculture and Human Values, 33, 439-454.
- Calloway, E. E., Carpenter, L. R., Gargano, T., Sharp, J. L., & Yaroch, A. L. (2023). New measures to assess the “Other” three pillars of food security–availability, utilization, and stability. International Journal of Behavioral Nutrition and Physical Activity, 20(1), 51.
- Cruz-Sánchez, Y., Aguilar-Estrada, A., Baca-del Moral, J., & Monterroso-Rivas, A. I. (2024). The availability of food in Mexico: an approach to measuring food security. Agriculture & Food Security, 13(1), 35.
- Martínez-Martínez, O. A., Gil-Vasquez, K., & Romero-González, M. B. (2023). Food insecurity and levels of marginalization: food accessibility, consumption and concern in Mexico. International Journal for Equity in Health, 22(1), 178.
RESPONSE:We appreciate your comment. In the revised version of the manuscript, the following changes and citations are presented. Lines: 137-141.
- Line 132: Please update the reference "[NO_PRINTED_FORM]" to the correct citation for the questionnaire validation.
RESPONSE:Updated reference. Line 219.
- Line 298: "On the other hand, although physical access and availability are adequate, they do not guarantee healthy eating or better food security status..." This sentence is a bit long. Consider breaking it into two or rephrasing for better flow.
RESPONSE:The visible changes have been added in lines: 464-466.
Reviewer 2 Report
Comments and Suggestions for Authors
The submitted paper makes a valuable contribution by examining food security among university students using both quantitative and qualitative data. However, several aspects of the paper require theoretical, methodological, and structural improvements. Below are my detailed comments and recommendations:
The abstract lacks basic methodological information. A brief mention of the approach (e.g., mixed-method, number of participants, data collection tools) would enhance clarity.
On page 2, lines 49–50, the authors mention that food security has often been evaluated through unidimensional approaches, but the manuscript fails to explain what these approaches entail and why they were preferred over multidimensional ones. A more detailed discussion is needed.
The rationale for selecting university students as the study sample is well-articulated and enhances the originality of the paper.
However, literature references (lines 64–71) are mostly from the American continent. Adding examples from European studies would enrich the context.
Although the dimensions of food security are discussed, the paper still lacks a clear explanation of what gaps in the literature it addresses and why the study is original. Strengthening the conceptual and theoretical framework is expected.
Under the Population and Sample section, only the quantitative sample is discussed. Why is the qualitative sample not included in this section? Integrating both would be more coherent.
The focus group section lacks important details: when and how was it conducted? Although purposive sampling was used to ensure representation from all faculties, it is unclear which faculties were included and based on what criteria.
How were the semi-structured interview questions developed? Was a pilot study conducted? Were they based on the literature?
The rationale for choosing the Chi-square test needs to be justified more clearly. Why was this test preferred over others?
Validity, reliability, and trustworthiness strategies for both quantitative and qualitative components should be described (e.g., triangulation, member checking, field notes).
The presentation of findings is quite complex. Quantitative and qualitative data (semi-structured interviews and focus groups) are presented together in a way that may confuse the reader. They should be presented in a more organized, thematically separated manner.
Theoretical implications are underdeveloped. Findings should be discussed in light of existing literature, and the paper’s contributions to theory must be stated more explicitly. The paper should clearly answer the question: What does this study add to the current body of knowledge?
Author Response
The submitted paper makes a valuable contribution by examining food security among university students using both quantitative and qualitative data. However, several aspects of the paper require theoretical, methodological, and structural improvements. Below are my detailed comments and recommendations:
The abstract lacks basic methodological information. A brief mention of the approach (e.g., mixed-method, number of participants, data collection tools) would enhance clarity.
RESPONSE: We thank you for your observation and add the visible changes in the lines: 23-28.
On page 2, lines 49–50, the authors mention that food security has often been evaluated through unidimensional approaches, but the manuscript fails to explain what these approaches entail and why they were preferred over multidimensional ones. A more detailed discussion is needed.
RESPONSE: The visible changes have been added in lines: 59-74.
The rationale for selecting university students as the study sample is well-articulated and enhances the originality of the paper.
However, literature references (lines 64–71) are mostly from the American continent. Adding examples from European studies would enrich the context.
RESPONSE: The visible changes have been added in lines: 90-96.
Although the dimensions of food security are discussed, the paper still lacks a clear explanation of what gaps in the literature it addresses and why the study is original. Strengthening the conceptual and theoretical framework is expected.
RESPONSE: The visible changes have been added in lines: 124-132.
Under the Population and Sample section, only the quantitative sample is discussed. Why is the qualitative sample not included in this section? Integrating both would be more coherent.
RESPONSE: The visible changes have been added in lines: 162-168.
The focus group section lacks important details: when and how was it conducted? Although purposive sampling was used to ensure representation from all faculties, it is unclear which faculties were included and based on what criteria.
RESPONSE: The visible changes have been added in lines: 251-259.
How were the semi-structured interview questions developed? Was a pilot study conducted? Were they based on the literature?
RESPONSE: The visible changes have been added in lines: 244-250.
The rationale for choosing the Chi-square test needs to be justified more clearly. Why was this test preferred over others?
RESPONSE: We appreciate the observation. The Chi-square test of independence was used because the variables analyzed were mostly categorical (nominal and ordinal), which precludes the use of parametric tests that require continuous data. Although normality was assessed for some variables, the nature of the data justifies the use of Chi-square as an appropriate statistical technique to determine associations between variables. This methodological decision is consistent with similar studies in the area of food safety.
In the revised version of the manuscript is mentioned as being appropriate for the categorical nature of the data. Lines: 267-268..
Validity, reliability, and trustworthiness strategies for both quantitative and qualitative components should be described (e.g., triangulation, member checking, field notes).
RESPONSE: The visible changes have been added in lines: 218, 231,232, 247, 248.
The presentation of findings is quite complex. Quantitative and qualitative data (semi-structured interviews and focus groups) are presented together in a way that may confuse the reader. They should be presented in a more organized, thematically separated manner.
RESPONSE: Changes were made to the results structure.
Theoretical implications are underdeveloped. Findings should be discussed in light of existing literature, and the paper’s contributions to theory must be stated more explicitly. The paper should clearly answer the question: What does this study add to the current body of knowledge?
RESPONSE: The visible changes have been added in lines: 489-497.
Reviewer 3 Report
Comments and Suggestions for Authors
This study addresses a relevant issue related to the identification of food security in the community college using the quantitative and qualitative approaches. However, I noticed points that need to be improved/clarified.
Abstract
- Insert more information about the methodological section, such as the year and place of data collection, the sampling method chosen, the statistical analysis and the level of significance adopted. Information on the questionnaires and instruments used.
- Present the statistical results found in the study such as proportions and p-values.
Introduction
- Insert reference on line 91.
- Specify the meaning of “integrated perspective” (line 99).
Methods
- Rethink the exclusion criteria and not the eligibility criteria among what has been presented (lines 115-116).
- “[NO_PRINTED_FORM]” I think it was a mistake (Line 132).
- Insert reference on line 150.
- Include a subsection on ethical aspects and describe the information presented on lines 153-154 and also insert this information related to the quantitative study.
- I suggest that the authors think about the possibility of building a methodological model and then presenting statistical models that show the association between food security and the exploratory variables investigated, controlling for possible confounding variables. These approaches could improve the study.
- Create a section on the analysis of food consumption explaining more about the instrument used, how this data was collected, the criteria used to group the foods mentioned by the participants and create a table describing all the food groups found in this analysis.
- Create another subsection describing all the exploratory variables investigated, not only with the instruments used to collect them, but also with the categorization adopted.
- Insert more information about the statistical analysis used, such as that described in the results section (Lines 210-212).
Results
In general, rethink the structure of the results section, as it is confusingly described.
- Insert the figure and/or table associated with the description of the result in every paragraph.
- Between lines 194-207 it is not clear where the results mentioned are described.
- Insert the proportions observed in this sample instead of just showing “lower presence”, “high consumption of ultra-processed foods” and so on (lines 194-207).
- Subsection 3.2: there is no need to repeat the information about the sample size, the p-value associated with normality and the terminology such as measures of central tendency and dispersion. This should be described in the method section.
- Was the dispersion value the standard deviation? Please, describe it in the method section.
- It is not recommended to use the symbol ± in reference to the standard deviation. Please replace it with SD in the text and tables.
- The results of the proportions described are not shown in table 1 (only the frequency) (rows 213-227).
- Please describe in more detail the associations observed in table 1.
- Was a post hoc test carried out to test the association in the case of variables with more than two categories? It is necessary to describe it.
- I suggest moving the last paragraph (lines 234-241) to the discussion section.
- Subsection 3.3: were the results presented between 243 and 246 observed in your study or do they describe the results of reference 23?
- Describe the acronym CONEVAL.
- In subsection 3.3, the association between food security and the exploratory variables described in table 1 could be described.
- Please present in table 2 all the proportions described in subsection 3.3. The text is not in line with table 2.
- Subsection 3.5: the information between 273-274 is related to the methods section.
Discussion
- First paragraph: highlight the numerical findings obtained in the study.
- Insert more quantitative results about the subject for comparison with the results found.
- Insert reference in the end on line 317.
- Describe the limitations and strengths of the study.
Tables and figures
- Improve the image quality of the figure.
- Table 1: indicate that within the brackets was present the proportion.
- Figure and tables: insert in the title information about the year, the country where the study was carried out and the sample size.
Author Response
This study addresses a relevant issue related to the identification of food security in the community college using the quantitative and qualitative approaches. However, I noticed points that need to be improved/clarified.
Abstract
- Insert more information about the methodological section, such as the year and place of data collection, the sampling method chosen, the statistical analysis and the level of significance adopted. Information on the questionnaires and instruments used.
RESPONSE The visible changes have been added in lines: 23-29.
- Present the statistical results found in the study such as proportions and p-values.
RESPONSE The visible changes have been added in lines: 29-34.
Introduction
- Insert reference on line 91.
- RESPONSE Reference added. Line 119.
- Specify the meaning of “integrated perspective” (line 99).
- RESPONSE The visible changes have been added in lines: 135-137.
Methods
- Rethink the exclusion criteria and not the eligibility criteria among what has been presented (lines 115-116).
- RESPONSE The visible changes have been added in lines: 159-161.
“[NO_PRINTED_FORM]” I think it was a mistake (Line 132).
RESPONSE Updated reference. Line 219.
- Insert reference on line 150.
- RESPONSE Updated reference. Line 163.
- Include a subsection on ethical aspects and describe the information presented on lines 153-154 and also insert this information related to the quantitative study.
- RESPONSE The visible changes have been added in lines: 276-280.
- I suggest that the authors think about the possibility of building a methodological model and then presenting statistical models that show the association between food security and the exploratory variables investigated, controlling for possible confounding variables. These approaches could improve the study.
- RESPONSE We are grateful for their valuable suggestions on the construction of advanced methodological and statistical models. Although we recognize the usefulness of multivariate models to control for possible confounding variables, in the present study we chose to perform bivariate analyses using chi-square tests to identify significant associations between food security and various sociodemographic variables. Additional correlations were also explored to identify possible relationships, although they were not adjusted for.
- Create a section on the analysis of food consumption explaining more about the instrument used, how this data was collected, the criteria used to group the foods mentioned by the participants and create a table describing all the food groups found in this analysis.
- RESPONSE The following were addressed in: 2.2.2 Variables on availability and physical access and 2.3.2.1 Structured observation of the food environment.
Lines: 180 y 234.
- Create another subsection describing all the exploratory variables investigated, not only with the instruments used to collect them, but also with the categorization adopted.
- RESPONSE The visible changes have been added in lines: 169-208.
- Insert more information about the statistical analysis used, such as that described in the results section (Lines 210-212).
- RESPONSE Additional information was added, lines 263-265. Additional qualitative analysis was included, lines 269-275.
Results
In general, rethink the structure of the results section, as it is confusingly described. RESPONSE Comment addressed.
- Insert the figure and/or table associated with the description of the result in every paragraph.
- RESPONSE
- Comment addressed.
- Between lines 194-207 it is not clear where the results mentioned are described. T
- RESPONSE this paragraph was rewritten to make it more precise.
- Insert the proportions observed in this sample instead of just showing “lower presence”, “high consumption of ultra-processed foods” and so on (lines 194-207).
- RESPONSE We thank you for your valuable observation. However, we clarify that the statements on the presence and frequency of available foods are derived from a qualitative assessment based on direct observation and semi-structured interviews with store managers, so exact quantitative proportions are not available. For clarity, we have clarified this limitation in the manuscript (see lines 355-364).
- Subsection 3.2: there is no need to repeat the information about the sample size, the p-value associated with normality and the terminology such as measures of central tendency and dispersion. This should be described in the method section.
- RESPONSE Comment addressed.
- Was the dispersion value the standard deviation? Please, describe it in the method section.
- RESPONSE The visible changes have been added in lines: 265.
- It is not recommended to use the symbol ± in reference to the standard deviation. Please replace it with SD in the text and tables.
- RESPONSE The visible changes have been added in lines: 265 y 328.
- The results of the proportions described are not shown in table 1 (only the frequency) (rows 213-227).
- RESPONSE Comment addressed. The proportions have been added to Table 1.
- Please describe in more detail the associations observed in table 1.
- RESPONSE The visible changes have been added in lines: 312-333
Was a post hoc test carried out to test the association in the case of variables with more than two categories? It is necessary to describe it.
RESPONSE We thank them for their comments. No post hoc tests were applied, as the analysis focused on identifying general associations between sociodemographic variables and food security status using the chi-square test. However, the patterns observed in Table 1 have been described in greater detail in the text review to facilitate interpretation of the associations.
- I suggest moving the last paragraph (lines 234-241) to the discussion section. RESPONSE Comment addressed. Moved to discussion, line 445-449
- Subsection 3.3: were the results presented between 243 and 246 observed in your study or do they describe the results of reference 23?
- RESPONSE We thank you for your comment. We confirm that the results presented in lines 243 to 246 correspond to findings obtained directly in our study, and not to reference 23. We regret the possible confusion. We made the necessary adjustments in the manuscript, eliminating this citation for clarity.
- Describe the acronym CONEVAL.
- RESPONSE The visible changes have been added in lines: 287.
In subsection 3.3, the association between food security and the exploratory variables described in table 1 could be described.
RESPONSE Thank you for your observation. We have complied with this request, and we have placed it in section 3.2 Sociodemographic characteristics and economic access, lines 312-333.
- Please present in table 2 all the proportions described in subsection 3.3. The text is not in line with table 2.
- RESPONSE Comment addressed. Table 2 was modified to match the proportions described in the section in question.
- Subsection 3.5: the information between 273-274 is related to the methods section.
- RESPONSE Comment addressed.
Discussion
- First paragraph: highlight the numerical findings obtained in the study. RESPONSE Comment addressed.
- Insert more quantitative results about the subject for comparison with the results found.
- RESPONSE The visible changes have been added in lines: 417-425, 438-447.
- Insert reference in the end on line 317.
- RESPONSE The paragraph was modified to improve the discussion on the food scholarship.
- Describe the limitations and strengths of the study.
- RESPONSE The visible changes have been added in lines: 498-519.
Tables and figures
- Improve the image quality of the figure.
- RESPONSE Comment addressed.
- Table 1: indicate that within the brackets was present the proportion. RESPONSE Comment addressed. They were indicated in the column headings (Table 1).
- Figure and tables: insert in the title information about the year, the country where the study was carried out and the sample size.
- RESPONSE Comment addressed.
Round 2
Reviewer 2 Report
Comments and Suggestions for Authors
Dear Authors
I would like to thank the authors for their thoughtful and thorough revisions. The methodological clarifications, theoretical enhancements, and the improved structure of the findings section have significantly strengthened the overall quality of the paper. In my view, the revised version adequately addresses the concerns raised in the initial review and represents a clear improvement over the original submission.
Author Response
Thank you so much
Reviewer 3 Report
Comments and Suggestions for Authors
I noticed points that still need to be improved/clarified.
Introduction section:
- Reduce the size of the section by summarizing the data presented.
Method section:
- Rethink the organization of the subsections of the methods section. I suggest that the description of the variables and how these data were analyzed be presented in the same subsection.
- To analyze categorical variables with more than two categories, it is necessary to apply a post hoc test.
Results section:
- Insert at the end of each paragraph the table/figure referring to the data described.
Author Response
- Reduce the size of the section by summarizing the data presented.
Attended (lines 73-88)
Method section:
- Rethink the organization of the subsections of the methods section. I suggest that the description of the variables and how these data were analyzed be presented in the same subsection.
Attended (lines 158-221)
- To analyze categorical variables with more than two categories, it is necessary to apply a post hoc test.
Attended (lines 73-88; 227-229; 294-301)
Results section:
- Insert at the end of each paragraph the table/figure referring to the data described.
Attended (Table 1)